# Association Between Renal Dysfunction and Lipid Ratios in Rural Black South Africans

**DOI:** 10.3390/ijerph22030324

**Published:** 2025-02-21

**Authors:** Cairo B. Ntimana, Reneilwe G. Mashaba, Kagiso P. Seakamela, Peter M. Mphekgwana, Rathani Nemuramba, Katlego Mothapo, Joseph Tlouyamma, Solomon S. R. Choma, Eric Maimela

**Affiliations:** 1DIMAMO Population Health Research Centre, University of Limpopo, Sovenga St, Polokwane 0727, South Africa; given.mashaba@ul.ac.za (R.G.M.); peacekagiso4@gmail.com (K.P.S.); witnessr84@gmail.com (R.N.); katlego.mothapo@ul.ac.za (K.M.); joseph.tlouyamma@ul.ac.za (J.T.); eric.maimela77@gmail.com (E.M.); 2Research Administration and Development, University of Limpopo, Sovenga St, Polokwane 0727, South Africa; peter.mphekgwana@ul.ac.za; 3Department of Pathology, University of Limpopo, Sovenga St, Polokwane 0727, South Africa; solomon.choma@ul.ac.za

**Keywords:** kidney dysfunction, AWI-Gen 1, lipids

## Abstract

In the past, it has been reported that the black South African population may have a cardio-protective lipid profile; however, this may no longer be the case with urbanization, industrialization, and the nutritional transition that occurred in South Africa. Although these transitions may be low in rural areas, one would expect this influence to be lower in the rural populations; however, they are not immune to these changes. Hence, the present study aimed to determine the association of serum lipid profiles and lipid ratios with kidney dysfunction. This cross-sectional retrospective study used the AWI-Gen 1 dataset. This study consisted of 1399 participants who took part in the AWI-Gen phase 1. Participants aged below 40 years, pregnant women, mentally disturbed and participants with incomplete information to answer the aims and objectives of this study were excluded in the analysis of this study. The data were analyzed using SPSS. In the present study, the prevalence of kidney dysfunction was 11.7%, with women having a significantly higher prevalence as compared to men. Women with kidney dysfunction had significantly higher TC, TG, TG/HDL-C, and TC/HDL-C compared to those without kidney dysfunction. However, in men, there was no association between the two groups. TC, and LDL/HDL-C were associated with kidney dysfunction in women only. TG/HDL-C was associated with kidney dysfunction in both women and men. Elevated TC, LDL/HDL-C, TC/HDL-C, and TG/HDL-C were the risk factors for kidney dysfunction, particularly in women. This suggests that TC, TC/HDL-C, and TG/HDL-C levels may be useful for risk stratification and a potential target to reduce the risk of developing kidney dysfunction, particularly in women. Upcoming longitudinal studies examining the causal connection between serum lipids, and lipid ratios with the risk of kidney dysfunction are necessary to fully understand the potential relationship between TG/HDL-C, TC, and TC/HDL-C levels and kidney dysfunction.

## 1. Introduction

Chronic kidney disease (CKD) has emerged as a major global public health issue, as its prevalence increases and so does the burden [1]. Worldwide, the cases of CKD were reported to be over 697 million in 2019 [2]. It is estimated to be the fifth leading cause of death globally by 2040 [3]. CKD increases the risk of infections [4] and cardiovascular risk factors [5], which prompts growing concern about a rising global mortality prevalence ranging from 8 to 16% [6]. Similarly, several studies have reported CKD to be a major public health concern in low- and middle-income countries (LMICs) [7]. The prevalence of CKD was reported to be at 11.2% in LMICs in Asia, with South Asian countries having the highest prevalence (13.5%) compared to East Asian countries, which have the lowest prevalence (8.6%) [8]. Approximately 10.7% of people in Sub-Saharan Africa had CKD in 2019, with South Africa accounting for a higher proportion (14%), compared to West African nations (6.6%) [9]. In South Africa, the fatalities caused by CKD increased by 67% from 1999 to 2006 [10]. These statistics underscore the urgent need for comprehensive research efforts to better understand CKD, its prevalence, associated risk factors, and its disproportionate impact on vulnerable populations. CKD is a progressive condition marked by structural and functional kidney impairments that result from complex interactions of hemodynamic, inflammatory, and metabolic mechanisms [1]. These include glomerular dysfunction, oxidative stress, dyslipidemia, and hypertension, which accelerate renal damage [9]. The condition is further aggravated by diabetes and mineral metabolism disturbances, leading to cardiovascular complications [11,12]. 

In South Africa, the prevalence of dyslipidemia was reported to range between 14% and 69%, with hypercholesterolemia affecting 28% of women and 19% of men in 2012 [11,12]. Moreover, 52% of men and 44% of women displayed low high-density lipoprotein cholesterol (HDL-C) levels. In addition, a study by Masilela et al. [13] reported a higher prevalence of dyslipidemia of 76.7%, with women having the highest prevalence as compared to men. Although the association between lipids and CKD has been documented in Western countries, there remains a gap in African populations. However, kidney dysfunction among rural black populations is reported to be a significant health concern, with risk factors including communicable and non-communicable diseases, which are more prevalent in women as compared to men [14].

CKD and cardiovascular disease (CVD) have some common risk factors, such as hypertension, obesity, and diabetes [1]. It is well known that serum lipids are associated with the development of CVD [15]. According to several studies, dyslipidemia, which is characterized by elevated levels of triglycerides (TGs), low-density lipoprotein cholesterol (LDL-C), and total cholesterol (TC), and reduced high-density lipoprotein cholesterol (HDL-C), is a risk factor for CVD [1,16]. Serum lipids might be associated with the development of CKD [6]. However, most studies reported contradicting findings, and the relationship between serum lipids and CKD remains controversial [6,17]. Several studies have reported a positive association between serum lipids and the onset of CKD [1,18,19], while other studies have found a negative association [20,21]. The inconsistencies between these may be due to the different study populations, and different ethnic groups.

In the past, it has been reported that the black South African population may have a cardio-protective lipid profile and that it was mainly the white population that had an adverse lipid profile. However, is this still the case with urbanization, industrialization, and the nutritional transition that occurred in South Africa? Although these transitions may be low in rural areas, one would expect this influence to be lower in the rural populations; however, they are not immune to these changes [22]. In addition, recent studies reported that African American patients have the fastest estimated glomerular filtration rate (eGFR) decline and are twice at risk of developing end-stage renal disease compared to other races [23,24,25,26]. Given that, the South African population may be at risk of renal disease due to the high prevalence of diabetes and hypertension, and there is a scarcity of data on whether serum lipids and lipid ratios are associated with CKD [16]. Moreover, there are no prior studies that have evaluated the association between lipids ratios and kidney dysfunction. Hence, the present study aimed to determine the prevalence of kidney dysfunction to confirm the sex differences in kidney dysfunction and to determine the association of serum lipid profiles and lipid ratios with kidney dysfunction to better understand this relationship, especially among the rural black population.

## 2. Materials and Methods

### 2.1. Study Design

This was a retrospective cross-sectional study. This study used Africa Wits INDEPTH-partnership for Genomic Research (AWI-Gen) phase one (1) data, which consisted of 1399 participants (428 men and 791 women) aged 40 and above. This study was conducted in the Dikgale, Mamabolo, and Mothiba Health Demographic Surveillance Site (DIMAMO HDSS), formally known as Dikgale Health Demographic Surveillance Site (Dikgale HDSS) in the Capricorn District of Limpopo Province, South Africa. The DIMAMO regions encompass rural and semi-urban areas within the HDSS. The predominant language spoken in this area is Northern Sotho. A significant portion of the population consists of black individuals, who are disproportionately represented among those with lower socioeconomic status and educational attainment. Moreover, the population density is 263 persons per kilometer square. The participants were selected using convenient sampling. Convenient sampling is a non-probability sampling technique where researchers select participants who are readily available and accessible to them [27]. AWI-Gen phase 1 study ran between 2014 and 2018. However, data were collected between 05 June 2014 and 11 October 2016. Informed consent was sought from the participants, and permission to use the data was sought from the principal investigator following this study’s data request protocol. Ethical clearance was sought from the University of Limpopo, Medunsa campus ethics committee (MREC) (MREC/HS/195/2014: CR). Permission to conduct this study in Dikgale villages was sought from the Dikgale Tribal Authority.

### 2.2. Selection Criteria

The present study included all participants who took part in AWI-Gen phase one (1) data collection. These were apparently healthy individuals aged 40 years and above. Participants aged below 40 years, pregnant women, and/or mentally disturbed were excluded from participating in this study. In addition, participants with incomplete information to answer the aims and objectives of this study (i.e., eGFR and ACR values, which were used to determine kidney dysfunction) were excluded from the analyses of the present study (see Figure 1). Approximately seven (7) participants were excluded. After excluding participants with missing data, the sample size of this study was 1392. More detailed methods on how the participants were recruited and how the data were collected are reported in the AWI-Gen study protocol [28]. Sampling bias could not be prevented as the primary study utilized convenience sampling in the AWI-Gen phase one (1) project. To prevent methodological bias, validated data collection tools and laboratory methods were employed. Statistical bias was prevented through the application of suitable statistical tests.

### 2.3. Measurements

Data were collected using an AWI-Gen questionnaire, and sociodemographic risk factors of CKD like age, sex, current alcohol consumption, and smoking were noted on the questionnaire. Biometric measurements were measured by qualified research assistants and included height, waist circumference, blood pressure, and body weight. With the participant wearing light clothing and no shoes, the Omron measuring scale made by Omron Healthcare was used to measure body weight to the nearest 0.1 kg (manufactured by Omron Healthcare Inc., Shanghai, China). Height was measured using a stadiometer; the participants were asked to stand vertically on the stadiometer without shoes and their body height was measured to the nearest 0.1 m. Body mass index (BMI) was determined by dividing the measured weight by the square of the height (kg/m^2^). Waist measurements were made using a measuring tape (SECA, Hamburg, Germany). Blood pressure was measured using an Omron blood pressure monitor (manufactured by Omron Healthcare Inc., Shanghai, China).

### 2.4. Serum Lipid Profiles and Lipid-Related Ratios

Overnight fasting blood was collected, and serum and plasma samples were analyzed in the chemical pathology laboratory, Department of Pathology and Medical Sciences, at the University of Limpopo. The Friedewald formula was used to calculate LDL-C, and total cholesterol, HDL-C, and triglycerides were measured enzymatically using Randox Daytona Plus clinical chemistry analyzer (UK) [29]. These were validated by a comparison of direct LDL-C measurements with Friedewald calculated values, and repeated measurements of the same sample over different days were carried out to compare the intra and inter coefficient of variation (≤5% was considered excellent precision), respectively [29].

Lipid ratios were used to predict the risk of CKD in the current study. The TG/HDL-C, TC/HDL-C, and LDL/HDL-C ratios were calculated by dividing TG, TC, and LDL by HDL-C. The normal cut-off values for serum lipid were TG level < 1.7 mmol/L, LDL-C level < 3.0 mmol/L, TC level < 5.00 mmol/L, and HDL-C > 1.0 mmol/L for men and >1.3 mmol/L for women [30]. The normal cut-off values for lipids ratios were as follows: TG/HDL-C < 4.7 for men and <3.7 for women, TC/HDL-C < 5.3, and LDL/HDL-C < 2.0 [31]. Dyslipidemia was defined as having serum total TC levels of 5.18 mmol/L or higher, TG levels of 1.7 mmol/L or higher, LDL-C levels of 3.0 mmol/L or higher, and low HDL-C levels of 1.0 mmol/L for men and 1.3 mmol/L for women [31].

### 2.5. Determination of CKD

Overnight morning urine samples were collected. Urinary albumin concentration was measured in the laboratory with immunoturbidimetry (Roche Cobas C501 System), and the concentration of creatinine was determined using an enzymatic method (Roche/Hitachi Cobas C501 System) and then used to determine the albumin-to-creatinine ratio (ACR) [28]. The glomerular filtration rate (GFR) is a measurement that allows for the estimation of renal clearance, which is the measurement of the clearance of a substance that is freely filtered by glomeruli and does not undergo reabsorption or tubular secretion, per volume of plasma filtered per unit of time (ml/min). The estimated glomerular filtration rate was calculated using the following equation: eGFR (mL/min/1.73 m^2^) = 141 × min (S-Cr/κ, 1) α × max (S-Cr/κ, 1) − 1.209 × 0.993 age × 1.018 [if female]. S-Cr is serum creatinine in µmol/L, κ is 61.9 for women and 79.6 for men, α is −0.329 for women and −0.411 for men. min indicates the minimum of S-Cr/κ or 1 and max indicates the maximum of S-Cr/κ or 1 [32,33]

The CKD-epi equation was used since it provides more accurate estimates of eGFR. However, the creatinine-based estimates of kidney function by GFR were not used in the present study since these are reported to be less accurate in certain populations, including patients with diabetes, pregnant women, and those with unusual body mass [34]. When estimating renal function with formulas based on creatinine, those with kidney impairment may not be correctly classified. While some people with underlying kidney illness may have relatively normal estimated GFR values, others may be diagnosed with renal dysfunction even when their actual kidney function is normal [34].

Due to the cross-sectional nature of this study, the data being collected once without follow-up testing to confirm the abnormalities, and this study being retrospective using secondary data, we were unable to meet the criteria for CKD and, instead, we used the term kidney dysfunction rather than CKD [35]. Kidney dysfunction was defined as a decreased eGFR of 60 mL/min/1.73 m^2^ and/or ACR ≥ 30 mg/mmol in the present study.

### 2.6. Determination of Obesity

Individuals with a body mass index (BMI) ≥ 30 kg/m^2^ were considered obese and those with a body mass index < 30 kg/m^2^ were considered non-obese [36,37]. The optimal cut-off values for WC are 94 cm in men and 80 cm in women. Central obesity/high waist circumference was defined as a waist circumference (WC) > 80 cm for women and >94 cm for men [38,39].

### 2.7. Determination of Hypertension

Normal adult blood pressure is defined as a systolic blood pressure (SBP) of less than 120 mmHg and diastolic blood pressure (DBP) of less than 80 mmHg [40]. Individuals with a history of hypertension, taking medication, or with blood pressure that is either >140 systolic mmHg or >90 diastolic mmHg, i.e., defined as 140/90 mmHg or above, were considered to be hypertensive [40].

### 2.8. Determination of Diabetes

Overnight fasting blood was collected and serum and plasma samples were analyzed in the chemical pathology laboratory. Glucose was determined using an AU480 auto-analyzer supplied by Beckman Coulter. Method performance: 40 µL of the reagent 1 and 20 µL of reagent 2 were mixed with 120 µL and 20 µL of diluent, respectively. The reaction mixture was amalgamated with 1.6 µL of the sample and incubated for 660 s at a wavelength of 340 nm. The normal fasting blood glucose level was <5.6 mmol/L [41]. Participants with a history of diabetes mellitus (DM), taking medication for diabetes, or with fasting glucose of ≥7.0 mmol/L or random glucose of ≥11.1 mmol/L were considered diabetic [42].

### 2.9. Data Analysis

#### 2.9.1. Exposure

The exposure of interest was kidney dysfunction (coded 0 = non-kidney dysfunction and 1 = kidney dysfunction).

#### 2.9.2. Outcome

The primary outcome of interest of the AWI-Gen 1 study was to evaluate kidney function among all participants. This was assessed using measures of eGFR and ACR. Kidney dysfunction was defined as an eGFR below 60 mL/min/1.73 m^2^ and/or an ACR equal to or greater than 30 mg/mmol. We rigorously excluded participants lacking eGFR and ACR data from our analysis to ensure data integrity. Among those with available eGFR and ACR data, we meticulously categorized them based on sex and their kidney dysfunction status. This allowed us to explore potential sex-specific associations between lipid profiles and lipid ratios and the incidence of kidney dysfunction. Our investigation explored whether particular lipid profiles and ratios were linked to the development of kidney dysfunction within distinct sex groups, offering valuable insights into the interplay between lipid metabolism and kidney health. To ensure the validity of the outcome, all laboratory biochemical measurements were performed according to good laboratory practice with external monitoring for quality control, as reported elsewhere [28].

#### 2.9.3. Covariates

Covariates included in the analysis were age at data collection (categorized as ≤45 years, 46 to 55 years, and ≥56 years), sex (men and women), body mass index (BMI), diabetes, hypertension, waist circumference, smoking, and current alcohol consumption.

#### 2.9.4. Statistical Analysis

Data were analyzed using statistical package for social sciences (SPSS) version 27.0 (I.B.M., Armonk, New York, NY, USA). Categorical variables were presented as percentages and continuous variables that were normally disturbed were presented as mean ± standard deviation, while those that were not normally distributed were presented as median (interquartile range). A comparison of proportions (i.e., kidney dysfunction vs. non-kidney dysfunction) was performed using chi-square, whilst a comparison of means was performed using an unpaired Student’s *t*-test. The relationship between kidney dysfunction and lipid profiles was determined by using bivariate correlation and partial correlation. The association between kidney dysfunction and lipid profiles was determined by binary logistic regression. Kidney dysfunction was coded “0” if the participants had elevated ACR and reduced eGFR, and coded “1” for non-kidney dysfunction if the participants had ACR and eGFR falling within the normal reference ranges. In the multivariate analysis, ACR and eGFR were the dependent variables. We adjusted for the determinants of kidney dysfunction, which included (age, sex, BMI, diabetes, hypertension, waist circumference, smoking, and current alcohol consumption); for all the inferential statistics, a *p*-value of less than 0.05 was considered statistically significant. In the present study, the relationship between serum lipid levels, lipid ratios, and kidney dysfunction was examined in terms of men and women.

## 3. Results

This study included 1392 participants, of which 30.7% were men and 69.3% were women. The mean age of participants was 52 ± 8.2 with no significant difference between men and women. About 11.7% of the participants had kidney dysfunction, with women having a significantly higher proportion (13.2 vs. 8.40, *p* = 0.011) as compared to men. The women had a significantly higher proportion of waist circumference, BMI, diastolic blood pressure, and hypertension (*p* = <0.001, <0.001, <0.001, and 0.008, respectively) than men. The proportion of current smokers and current alcohol consumption was significantly higher in men than in women (*p* = <0.001, and <0.001, respectively). Women had significantly higher levels of total cholesterol, LDL-C, and TC/HDL-C ratios than men. Men had significantly higher HDL-C levels, eGFR, and serum creatinine as compared to women (refer to Appendix A).

Table 1 presents a comparison of the anthropometric and biochemical characteristics between participants with kidney dysfunction and those without kidney dysfunction. The results were categorized by sex to control the effect of sex on associations. Men with kidney dysfunction were significantly older, with higher SBP, DBP, and higher glucose levels as compared to men without kidney dysfunction, respectively (56.75 ± 9.68 vs. 51.50 ± 8.03, *p* = 0.003), (140.24 ± 2.4 vs. 124.76 ± 20.36, *p* = 0.002), (84.90 ± 13.86 vs. 78.09 ± 12.55, *p* = 0.007) (6.51 ± 4.78 vs. 4.91 ± 1.56, *p* = 0.053). Men with kidney dysfunction had a higher prevalence of diabetes mellitus and hypertension compared to those without kidney dysfunction, respectively (16.7% vs. 4.7%, *p* = 0.011), (50.0% vs. 21.5%, *p* = 0.001). Men without kidney dysfunction had a higher proportion of smoking as compared to those with kidney dysfunction (63.2% vs. 27.8%, *p* = <0.001). However, amongst women, there was no significant difference between the two groups. There was no significant difference in current alcohol consumption between both men and women with kidney dysfunction and those without kidney dysfunction. There was no significant difference in serum lipids and their ratios between men with kidney dysfunction and those without kidney dysfunction. In contrast to men, women with kidney dysfunction had significantly higher TC, TG, TG/HDL-C, and TC/HDL-C compared to those without kidney dysfunction (*p* = 0.050, 0.003, <0.001, and 0.003, respectively). There was no significant difference in LDC-C/HDL-C, HDL-C, and LDL-C between women with kidney dysfunction and those without kidney dysfunction.

Figure 2 illustrates the binary logistic regression between kidney dysfunction and lipids profiles. In women, a TG/HDL-C ratio of 2.089 (aOR = 2.089; 95%CI: 0.378–11.556) was more likely associated with kidney dysfunction, while no significant associations were observed between kidney dysfunction and LDL-C, HDL-C, TG, or TC/HDL-C. Similarly, in men, no significant associations were found between kidney dysfunction and LDL-C, HDL-C, TG, TC/HDL-C, or LDL/HDL-C. However, in men, an elevated TG/HDL-C ratio of 2.028 (aOR = 2.028; 95%CI: 1.158–26.064) was significantly linked to kidney dysfunction.

In bivariate correlation, there was no significant association between lipids (individual lipids and lipid rations), eGFR, and ACR in men. The same association was also noted in women, whereby there was no association between ACR and lipid profiles. However, TC, TG, TC/HDL-C, and TG/HDL-C correlated positively and significantly with eGFR in women (Table 2).

In partial correlation, there was no association between ACR and lipids in both sexes after adjusting for age, hypertension, general obesity, diabetes, waist circumference, current alcohol consumption, and current smoking. TG/HDL-C correlated positively and significantly with eGFR in men, with other lipids profiles not being statistically significant. TC correlated positively and significantly with eGFR in women (Table 3). TG, DL-C, HDL-C, and the lipid ratios were not statistically significant.

In multivariate analysis, after adjusting for age, and other covariates in women, eGFR correlated positively with TG, TG/HDL-C, and TC/HDL-C, TC, whereas, among men, that was not the case. There was no association between ACR and eGFR with LDL-C in both men and women. However, HDL-C did have a positive correlation with ACR in men but not significantly in women. In both sexes, there was no relationship between ACR and lipid ratios (Table 4).

## 4. Discussion

This study aimed to determine the association between lipid profiles and kidney dysfunction. In the present study, the prevalence of kidney dysfunction was 11.7%. In contrast with the present study, Matsha et al. [9], in a study conducted in the urban Western Cape, South Africa, reported a higher prevalence of CKD (17.3%). Similarly, a study by Navise et al. [1] reported a higher prevalence of kidney dysfunction of 29.2% in a population of urban and rural areas. The inconsistencies between the findings of the present study and those of Navise et al. [1] and Matsha et al. [9] may be due to the following: The study by Matsha et al. [43] was conducted amongst participants of mixed ancestry (mixed race) therefore making the results not translatable to the black rural population. The study by Navise et al. [1] was conducted amongst HIV participants, which might have increased the prevalence of kidney dysfunction as other studies have documented CKD to be more prevalent in HIV individuals as compared to the general population [44,45]. In addition, the differences in demographics, health status, and study designs may contribute to the variation in the prevalence of kidney dysfunction observed [46].

Significantly more women had kidney dysfunction when compared to men. In agreement with the present study, Bouya et al. [47] reported similar findings, with women having the highest prevalence of CKD as compared to men. The reason for kidney dysfunction to be more prevalent in women as compared to men may be the negative impact of estrogen on triglycerides, which can cause kidney dysfunction [16]. Studies have shown that estrogen can influence lipid metabolism, but the relationship with kidney health is complex and multifactorial [48]. After adjusting for hypertension, general obesity, diabetes, waist circumference, current alcohol consumption, and current smoking, the association between lipids profiles remained more significant in women than in men, thus suggesting that other factors such as hormonal differences play a role. In addition, women are more likely to develop kidney damage as a result of pregnancy-related issues like high blood pressure [48]. Diabetes mellitus [48], obesity [5,49], central obesity [50], and hypertension [10,43] are reported to be the main contributing risk factors of kidney disease and, in the present study, they were more prevalent in women as compared to men, which could be the reason why kidney dysfunction was more prevalent in women as compared to men.

The findings of this study further indicated that both men and women experiencing kidney dysfunction were notably older, with elevated systolic blood pressure (SBP), diastolic blood pressure (DBP), and increased glucose levels compared to those without kidney dysfunction. In agreement with the findings of the present study, Navice et at. [1] indicated that, compared to those with normal kidney function, participants with eGFR < 90 mL/min/1.73 m^2^ and/or uACR ≥ 3.0 mg/mmol tended to be older, female, and exhibited greater levels of adiposity, as well as higher systolic and diastolic measures. Given the considerable prevalence of hypertension and diabetes mellitus in South Africa [51,52], it is understandable that we observed individuals with kidney dysfunction to have elevated an SBP in our population considering the association between CKD and elevated blood pressure and diabetes mellitus [10,53,54].

Women had a higher mean of TC, LDL-C, and TC/HDL-C as compared to men. However, the mean HDL-C was significantly higher in men as compared to women. In agreement with the present study, a study by Ringane and Choma [55] reported similar findings. Similarly, several studies have indicated that women typically demonstrate elevated TC, LDL-C, and TC/HDL-C ratios, whereas men frequently display increased HDL-C levels [55,56]. In the present study, among men, there were no notable variances in serum lipids and their ratios between those with kidney dysfunction and those without. Conversely, women with kidney dysfunction had significantly higher TC, TG, and TG/HDL-C ratios compared to women without kidney dysfunction.

Previous studies conducted in South Africa (HEART of SOWETO study, SABPA study, Agincourt and Free State study) reported that the black South African population may have a cardio-protective lipid profile and that it was mainly the white population that had an adverse lipid profile. However, due to urbanization, industrialization, and the nutritional transition that is occurring in South Africa, the relevance of lipids profiles in the black population and how these contribute to CVD and CKD may have changed; hence, they should be considered one of the main risk factors when investigating renal dysfunction [57,58,59]. Although these transitions may be low in rural areas, one would expect this influence to be lower in the rural populations; however, they are not immune to these changes [22]. The present study found that TG/HDL-C was associated with an increased risk of kidney dysfunction in both sexes. In binary logistic regression, TG/HDL-C was associated with kidney dysfunction in both women and men. Previous studies have reported TG/HDL-C to play a significant role in the pathogenesis and progression of kidney dysfunction [57,60,61]. A cohort study by Weldegiorgis and Woodward [16] reported TG/HDL-C to be the independent risk factor for CKD. Increased TG/HDL-C may be linked to the kidney’s glomeruli hardening, which signifies the onset of CKD [62]. Although the pathogenesis of CKD by TG/HDL-C is not clearly understood [16], one suggested mechanism is that elevated TG/HDL-C stimulates glomerular mesangial cells, which secrete pro-inflammatory cytokines such as interleukin-6, tumor necrosis factor-α, and transforming growth factor-β, which can damage the glomeruli and ultimately result in the development of glomerulosclerosis [62]. Furthermore, increased TG/HDL-C increases the levels of small dense LDL-C particles, which are reported to the highly atherogenic [63,64]. An increase in the blood concentration of small dense LDL-C and chylomicron remnants stimulates monocytes and macrophages to release pro-inflammatory cytokines and chemokines and increases inflammation and oxidative stress [62]. Oxidative stress and inflammation promote a decline in eGFR [64].

There was a significant association between TC, TG, LDL/HDL-C, and TC/HDL-C and kidney dysfunction in women but not in men in the findings of the present study. However, a study by Wu et al. [65], in a Chinese population, reported TC/HDL-C to be associated with CKD in both women and men. The reasons for the difference between the present study and the previous study may be that the latter was conducted in a developed country, urban setting, while the present study was conducted in a rural setting, and different ethnic groups. A study by Navise et al. [1], in a mixed urban and rural population in the North West, reported an elevated LDL-C/HDL-C ratio to be associated with decreased eGFR. However, this is in contradiction to our results as we observed the association only in women. The differences between the present study and the study by Navise may be due to different study settings and that the latter also considered individuals living with HIV. Moreover, the study only considered the LDL-C/HDL-C ratio, whereas the present study focused on all the serum lipid profiles.

In the present study, we observed lipid profiles to be associated with kidney dysfunction, mainly in women as compared to men. This may be attributed to the sample size and sex inequality. In addition, this study was conducted in adults, with a mean age of 52 years. Women in this age group are more likely to be menopausal or premenopausal. Menopausal women are reported to have elevated estrogen, which may have a negative impact on lipid profiles and ultimately renal dysfunction [57]. These results underscore the significance of monitoring lipid profiles in women experiencing kidney dysfunction as these ratios can offer valuable indications of cardiovascular health and overall risk evaluation. It is crucial to address modifiable risk factors such as dyslipidemia and blood pressure management in this demographic.

### Study Limitations

The findings of this study cannot be generalized to other areas in South Africa since this site is a very small rural area in one specific province and conditions may differ significantly in urban areas or among different racial populations. Due to the cross-sectional study design, we could not establish the causality or temporality of events, which requires a longitudinal study design. Thus, this study included a single measurement of eGFR and ACR without confirmation after 3 months, which prohibited the confirmation of the CKD. The authors acknowledge that CKD can affect lipid profiles. Thus, whether dyslipidemia caused the increase in CKD cannot be determined in this cross-sectional study. Nevertheless, we believe that the present study provided insight into the association between kidney dysfunction, serum lipids, and lipid ratios among the rural black population.

## 5. Conclusions

The prevalence of kidney dysfunction was 11.7%. Kidney dysfunction was associated with TC, LDL/HDL-C, and TC/HDL-C in women. However, TG/HDL-C is associated with kidney dysfunction in both women and men. This suggests that TC, TC/HDL-C, and TG/HDL-C levels may be useful for risk stratification and a potential target to reduce the risk of developing kidney dysfunction, particularly in women. Upcoming studies examining the causal connection between serum lipids and lipid ratios with the risk of kidney dysfunction are necessary to fully understand the potential relationship between TG/HDL-C, TC, and TC/HDL-C levels and kidney dysfunction. Furthermore, the incorporation of the findings of the present study in designing policy for clinical practice may positively contribute to the management of CKD in rural black communities.

## Figures and Tables

**Figure 1 ijerph-22-00324-f001:**
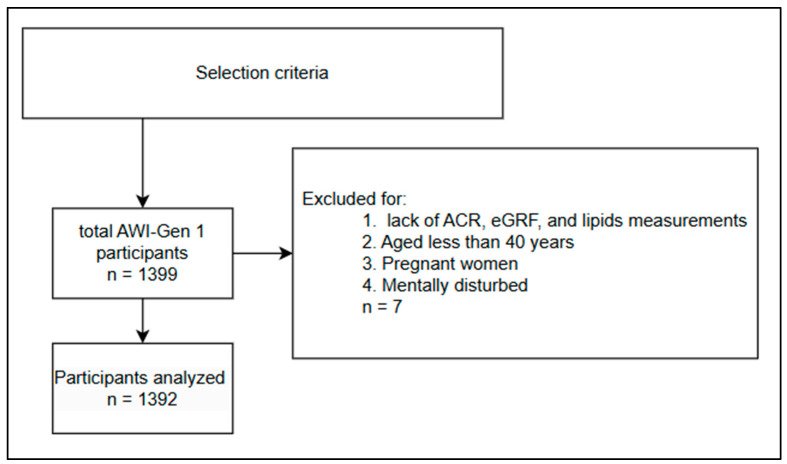
Selection criteria.

**Figure 2 ijerph-22-00324-f002:**
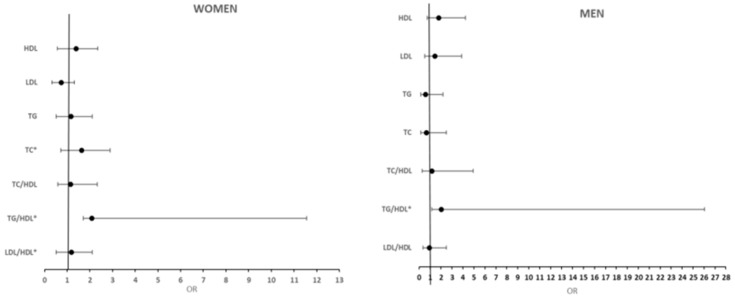
Figure plot illustrating the binary logistic regression of kidney dysfunction and lipid and lipids rations. Normal = 1, OR > 1 = positive relationship, OR < 1 = negative relationship. * *p* = <0.05.

**Table 1 ijerph-22-00324-t001:** Comparison of kidney dysfunction and non-kidney dysfunction with risk factors.

	Men			Women		
Variables	Non-Kidney Dysfunction (N = 391)	Kidney Dysfunction (N = 36)	*p*-Value	Non-Kidney Dysfunction (N = 838)	Kidney Dysfunction (N = 127)	*p*-Value
Age (mean ± SD)	51.50 ± 8.03	56.75 ± 9.68	0.003	51.78 ± 8.13	54.4 ± 8.64	0.001
Age (≤45 yrs) n (%)	104 (26.6)	6 (16.7)	0.008	208 (24.8)	19 (15.0)	0.020
Age (46–55 yrs) n (%)	169 (43.2)	10 (27.8)		352 (42.0)	53 (41.7)	
Age (≥56 yrs) n (%)	118 (30.2)	20 (55.6)		278 (33.2)	55 (43.3)	
BMI (kg/m^2^)	21.57 ± 3.92	23.31 ± 5.02	0.051	30.55 ± 7.98	30.3 ± 7.26	0.740
Obesity n (%)	11 (2.8)	3 (8.3)	0.105	420 (50.1)	58 (45.7)	0.392
WC (cm)	80.10 ± 11.29	86.4 ± 13.55	0.010	93.58 ± 15.91	95.62 ± 17.06	0.205
Central obesity n (%)	50 (12.8)	10 (27.8)	0.022	653 (78.0)	102 (80.3)	0.644
SBP (mmHg)	124.76 ± 20.36	140.24 ± 27.74	0.002	125.3 ± 20.63	132.2 ± 26.03	0.005
DBP (mmHg)	78.09 ± 12.55	84.90 ± 13.86	0.007	81.14 ± 12.43	85.14 ± 15.21	0.004
Hypertension n (%)	84 (21.5)	18 (50.0)	<0.001	246 (29.4)	51 (40.2)	0.017
Current smoker n (%)	247 (63.2)	10 (27.8)	<0.001	27 (3.2)	4 (3.1)	1.000
Current alcohol consumption n (%)	237 (60.8)	19 (52.8)	0.377	114 (13.6)	16 (12.6)	0.889
Glucose (mmol/L)	4.91 ± 1.56	6.51 ± 4.78	0.053	5.22 ± 2.22	6.07 ± 3.57	0.014
Diabetes mellitus n (%)	18 (4.7)	6 (16.7)	0.011	18 (15.1)	52 (6.3)	0.002
TC (mmol/L)	3.95 ± 1.03	3.95 ± 1.04	0.994	4.19 ± 1.09	4.49 ± 1.68	0.050
TG (mmol/L)	0.95 (1.34–0.682)	0.99 (1.29–0.73)	0.785	0.95 (1.32–0.71)	1.15 (1.51–0.78)	0.003
LDL-C (mmol/L)	2.36 ± 1.03	2.27 ± 0.95	0.614	2.64 ± 1.02	2.59 ± 1.01	0.549
HDL-C (mmol/L)	1.26 ± 0.47	1.20 ± 0.57	0.524	1.19 ± 0.36	1.13 ± 0.34	0.117
TG/HDL-C	0.83 (1.27–0.53)	0.73 (1.38–0.58)	0.725	0.84 (1.28–0.56)	1.03 (1.45–0.78)	<0.001
TC/HDL-C	3.51 ± 1.68	3.72 ± 1.52	0.430	3.74 ± 1.16	4.14 ± 1.41	0.003
LDL-C/HDL-C	2.22 ± 2.7	2.16 ± 1.06	0.808	2.39 ± 1.03	2.47 ± 1.11	0.494

**Table 2 ijerph-22-00324-t002:** Bivariate correlation of ACR and eGFR and lipids.

	Men				Women			
	eGFR		ACR		eGFR		ACR	
Variables	Correlation	*p*-Value	Correlation	*p*-Value	Correlation	*p*-Value	Correlation	*p*-Value
HDL-C (mmol/L)	−0.054	0.266	−0.085	0.265	0.006	0.849	−0.044	0.399
LDL-C (mmol/L)	−0.044	0.370	0.006	0.940	−0.022	0.515	−0.045	0.398
TC (mmol/L)	0.063	0.196	−0.020	0.798	0.142	<0.001	0.054	0.296
TG (mmol/L)	0.054	0.267	0.024	0.754	0.108	0.001	0.032	0.534
LDL/HDL-C	−0.008	0.868	0.085	0.272	−0.031	0.353	0.010	0.846
TC/HDL-C	0.089	0.065	0.051	0.503	0.075	0.020	0.092	0.074
TG/HDL-C	0.081	0.097	0.053	0.488	0.069	0.032	0.071	0.169

**Table 3 ijerph-22-00324-t003:** Partial correlation of ACR and eGFR and lipids.

	Men				Women			
	eGFR		ACR		eGFR		ACR	
Variables	Correlation	*p*-Value	Correlation	*p*-Value	Correlation	*p*-Value	Correlation	*p*-Value
HDL-C (mmol/L)	−0.013	0.801	−0.085	0.265	−0.028	0.394	−0.080	0.140
LDL-C (mmol/L)	−0.049	0.325	0.006	0.940	−0.049	0.139	−0.043	0.429
TC (mmol/L)	0.049	0.322	−0.020	0.798	0.067	0.046	−0.026	0.633
TG (mmol/L)	0.048	0.333	0.024	0.754	0.052	0.122	0.042	0.436
LDL/HDL-C	−0.010	0.840	0.085	0.272	−0.045	0.180	0.024	0.656
TC/HDL-C	0.069	0.165	0.051	0.503	0.048	0.148	0.080	0.139
TG/HDL-C	0.105	0.035	0.053	0.488	0.033	0.319	0.087	0.168

Adjusted for age, hypertension, diabetes, waist circumference, current alcohol consumption, and current smoking.

**Table 4 ijerph-22-00324-t004:** Multivariate logistic regression of eGFR and ACR with lipids profiles.

	Men				Women			
	eGFR		ACR		eGFR		ACR	
Variables	OR (95%CI)	*p*-Value	OR (95%CI)	*p*-Value	OR (95%CI)	*p*-Value	OR (95%CI)	*p*-Value
TC	1.47 (0.83;2.69)	0.190	0.95 (0.66;1.38)	0.79	1.55 (1.23;1.94)	<0.001	1.10 (0.92;1.32)	0.30
HDL	2.41 (0.58;10.0)	0.23	2.79 (1.19;6.53)	0.018	1.03 (0.50;2.11)	0.94	1.31 (0.70;2.45)	0.39
LDL	0.44 (0.52;3.67)	0.45	2.05 (0.83;5.06)	0.12	0.66 (0.33;1.29)	0.23	0.98 (0.59;1.65)	0.95
TG	1.00 (0.12;8.67)	0.99	0.60 (0.15;2.44)	0.48	2.75 (1.32;3.75)	0.007	0.98 (0.49;1.99)	0.98
TC/HDL-C	1.20 (0.97;1.49)	0.86	1.06 (0.89;1.26)	0.50	1.26 (1.03;1.53)	0.021	1.17 (0.98;1.40)	0.08
TG/HDL-C	1.22 (0.94;1.59)	0.141	1.10 (0.83;1.46)	0.49	1.41 (1.03;1.94)	0.035	1.24 (0.91;1.69)	0.17
LDL/HDL-C	0.96 (0.57;1.61)	0.87	1.21 (0.86;1.72)	0.27	0.86 (0.64;1.18)	0.35	1.02 (0.82;1.27)	0.85

Adjusted for age, hypertension, general obesity, diabetes, waist circumference, current alcohol consumption, and current smoking.

## Data Availability

The data presented in this study are available on request from the corresponding author. The data are not publicly available due to privacy or ethical restrictions.

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
