# Peer review of "Association Between Renal Dysfunction and Lipid Ratios in Rural Black South Africans"

_ijerph, 2025, doi:10.3390/ijerph22030324_

Round 1
Reviewer 1 Report
Comments and Suggestions for Authors
Evaluation of the manuscript “The Association Between Renal Dysfunction and Lipid/Lipid Ratios Among the Rural Black Population of South Africa,” submitted to the International Journal of Environmental Research and Public Health
This study examines the relationship between lipid profiles and renal dysfunction in a rural South African population, a topic that has been minimally explored in the literature. The methodology is appropriate, and the results are well-structured, making the study relevant to the fields of public health and nephrology. However, some revisions are needed to enhance clarity.
The title is clear and informative, effectively reflecting the study’s scope and key factors of analysis (renal dysfunction, lipids, and the rural South African population). However, it could be more concise and avoid redundancies. Suggested revision: “Association Between Renal Dysfunction and Lipid Ratios in Rural Black South Africans.”
The abstract provides a well-structured overview of the study, including its context, objectives, methodology, key results, and conclusions. However, the methods section could offer more details regarding the sample and inclusion/exclusion criteria. Additionally, the results section should further emphasize the practical implications of the findings, particularly the differences between men and women in the relationship between lipid profiles and kidney dysfunction.
The introduction effectively contextualizes the problem, highlighting the significance of chronic kidney disease and its association with lipid profiles. The literature review is well integrated to support the study. However, the authors should make the research justification more explicit, directly linking the gap in the literature to the necessity of this study within the rural Black population of South Africa.
The study is well-designed as a retrospective cross-sectional analysis, utilizing a large dataset (AWI-Gen Phase 1) with 1,392 participants. The laboratory and statistical methods are appropriate for the analysis, and the authors acknowledge the limitation of convenience sampling. Details regarding participant recruitment are provided in the AWI-Gen study protocol, as referenced by the authors.
The results are presented clearly, with well-organized means, standard deviations, and percentages. The correlations are thoroughly detailed, illustrating how different lipid profiles are associated with renal dysfunction, particularly among women. Some findings are not statistically significant, and their interpretation could be further expanded to prevent potential misinterpretations.
The discussion effectively connects the results to the existing literature, explaining both similarities and differences in comparison to previous studies. The impact of gender differences on renal dysfunction and lipid profiles is well addressed.
The conclusion should place greater emphasis on the study’s impact on clinical practice and public health policy. Strengthening this section would enhance the study’s relevance and its applicability in healthcare.
Author Response
Dear reviewer
Thanks for the comments.
Please find the attached table of corrections

Reviewer 2 Report
Comments and Suggestions for Authors
Dear all,
Comments follow throughout the document.

Author Response
Dear Reviewer.
Thanks for the comments
Please find the attached table of corrections

Reviewer 3 Report
Comments and Suggestions for Authors
This article addresses the association between renal dysfunction and Lipid/Lipid ratios amongst the rural black population of South Africa. The topic is relevant, but major deficiencies identified in both content and form need to be addressed based on the specific recommendations below:
- The conclusion part of the abstract should be improved in terms of outcomes and the future research directions this study may refer to.
- Multiple bibliographic citations (i.e. [1-3], [6-8] etc.) can increase redundancy and decrease the correlation of the information presented with the references indicated. Either make the information more specific and divide the references, or better yet, remove some of them.
- L52 - [14,15] bibliographic indexes are per se structures of the article and are not linked to any word in the text. Please revise the whole manuscript from this point of view.
- The design of the study subsection and the inclusion and exclusion criteria in the study should be better organized and detailed for better understanding, so presenting them in a diagram would perhaps improve understanding of the study design.
- Sections and subsections should be numbered as in the example provided as a journal template. Likewise, the titles of tables should be formatted according to the journal’s guidelines. The authors should carefully review the instructions for authors.
- L321, 337, and 348 repeat the same expression ‘in the present study’. Please review and modify accordingly.
- To add value to the manuscript and indicate that a variety of factors may contribute to renal imbalances, reference should be made to exploring the potential links between environmental exposures and renal dysfunction. This could offer a more comprehensive perspective on the factors influencing renal health in rural populations. I suggest checking and referring to: https://doi.org/10.1016/j.etap.2024.104620 and PMID: 39566262.
- The iThenticate report indicates a similarity of 29%. This should be significantly reduced.
Author Response
Dear Reviewer
Thanks for the comments
Please find the attached table of corrections

Round 2
Reviewer 2 Report
Comments and Suggestions for Authors
Dear all,
Once again I suggest that the introduction should have a more specific approach to the characterization of CKD in relation to its pathophysiology.
Kind regards,
Author Response
Dear reviewer
Thanks for the comments
Please find the attached tables of comments

Reviewer 3 Report
Comments and Suggestions for Authors
The authors have significantly improved the manuscript.
Author Response
Dear reviewer
Thank you